# Brief Communication: A low cost Arduino®-based wire extensometer for earth flow monitoring

Luigi Guerriero[1], Giovanni Guerriero[2], Gerardo Grelle[1], Francesco M. Guadagno[1], Paola Revellino[1]

[1]Department of Science and Technology, University of Sannio, 11 Port'Arsa street, 82100, Benevento, Italy

[2]Independent researcher, 5 Accesso alla Nazionale street, 83010, Capriglia Irpina, Avellino, Italy

*Correspondence to*: Luigi Guerriero (luigi.guerriero@unisannio.it)

**Abstract.** Continuous monitoring of earth flow displacement is essential for the understanding of the dynamic of the process, its ongoing evolution and to design mitigation measures. Despite its importance, it is not always applicable due to its expensiveness, and the need of integration with additional sensors for monitoring factors controlling movement. To overcome these problems, we developed and tested a low-cost Arduino-based wire-rail extensometer integrating a datalogger, a power system and multiple digital and analog inputs. The system is equipped with a high precision position transducer that in the test configuration offers a measuring range of 1023 mm and an associate accuracy of ±1mm and integrates an operating temperature sensor that should allow to identify and correct potential thermal drift that typically affects this kind of systems. Field test, conducted at the Pietrafitta earth flow where additional monitoring systems had been installed, indicates a high reliability of the measurement and a high monitoring stability without visible thermal drift.

## 1 Introduction

Earth flows activity alternates between long periods of slow and/or localized movements and surging events (e.g. Guerriero et al., 2015). Slow movement is normally concentrated along lateral slip-surfaces (Fleming and Johnson, 1989; Gomberg et al., 1995; Coe et al., 2003; Guerriero et al., 2016) which consists of fault-like segments (e.g. Segall and Pollard 1980) locally associated with cracks arranged in en echelon sets (Fleming and Johnson, 1989). Movement velocity is controlled by hydrologic forcing and seasonal acceleration and deceleration are induced by variation of the pore-water pressure (e.g. Iverson, 2005; Grelle et al., 2014). Thus, most earth flows move faster during periods of high precipitation or snowmelt than during drier periods, and the correlation between precipitation and velocity are normally complex (Coe et al., 2003; Schulz et al., 2009). Earth flow surge can occur when prolonged rainfalls are associated with the loss of an efficient drainage pathways (Handwerger et al., 2013) and new sediment becomes available in the source zone through retrogression of the upper boundary (Guerriero et al., 2014). In these conditions, the earth flow material can fluidize and fails catastrophically.

Each different kinematic behavior materializes a specific hazard level that needs to be quantified on the basis of monitoring data. An accurate identification of hazard, includes also the understanding of factors controlling earth flow movement (Shultz et al., 2009). In this way, a continuous record of earth flow displacement and its environmental drivers is essential in

defining the dynamic of the process (e.g. Corominas et al., 2000). Additionally, for earth flow involving human infrastructures (e.g. roads and railroads) displacement monitoring is crucial to understand the ongoing evolution and to design mitigation measures.

Displacement monitoring can be completed with a variety of instrumentations (e.g. rTPS, GPS etc…), but most of them i) does not allow nearly continuous monitoring, ii) implies time-consuming and expensive monitoring campaigns and/or iii) cannot be integrated with additional sensors (Corominas et al., 2000). Wire extensometers are particularly suitable for continuous monitoring, especially when it is concentrated along well-defined shear surfaces, and can be easily integrated in multi-sensors monitoring systems. Major disadvantages of extensometers are their cost (a single point high performance sensor is sold at ~1000€) and their sensitivity to temperature that is also function of the characteristics of cabling system. In this paper, we present a new Arduino®-based wire-rail extensometer specifically developed for monitoring of earth flow movement. We chose the Arduino board because it has been successfully used for the development of monitoring systems for different applications (e.g. Bitella et al., 2014; Di Gennaro et al., 2014; Lockridge et al., 2016). The system integrates a power unit, a datalogger, and an operating temperature sensor, has a very low cost (~200€), is configurable with different measurement ranges and accuracy, and has the potential to work with additional sensors. We test extensometer performance at the Pietrafitta earth flow in southern Italy comparing its measurement with those derived by successive GPS surveys and discrete rTPS measurements.

## 2 Extensometer components, structure, and code logic

The extensometer is composed of a i) processing and storage modules, ii) an onboard operating temperature sensor, iii) a linear position transducer, and iv) a power unit (Fig. 1; a simplified assemblage guide is reported in the supplemental). The processing and storage modules are an Arduino Uno board and a XD-05 Datalogging shield, respectively. The Arduino Uno board is a user-friendly version of an integrated microcontroller circuit that uses an ATmega238p low-power CMOS 8-bit microcontroller. It has 14 digital input/output pins, 6 analog inputs, a 16 MHz quartz crystal, a USB connection, a power jack, an ICSP header and a reset button (https://www.arduino.cc). The presence of multiple digital and analogs inputs make this platform ready for reading multiple sensors. The datalogging shield integrates a SD card read/write slot, a Real Time Clock (RTC) module with coin cell battery backup and a prototyping area. In this way, monitoring data are logged in a SD card at predefined interval and date/time is associated. To choose this shield, we considered the presence of the RTC module and its cost. We choose the cheapest. The operating temperature sensor consists of a 10K thermistor characterized by a tolerance of ±5%. It is installed with a reference resistor of 10K in the prototyping area of the logging shield (see wiring schematic in the code attached as supporting material). For our test, the thermistor was calibrated between -10 and 40°C and obtained Steinhart-Hart coefficients were used for temperature estimation. Steinhart-Hart coefficients were calculated using SRS Thermistor Calculator (http://www.thinksrs.com/downloads/programs/Therm%20Calc/NTCCalibrator/NTCcalculator.htm). We estimated temperature error comparing thermistor measurement with a precision thermometer (accuracy ±0.05°C) in laboratory

controlled conditions. The RMSE calculated on the basis of 40 observations (between -5 and 35°C) was of ~1°C. The linear position transducer is responsible for measuring cumulated displacement and consists of a 1Kohms Bourns 3540S-1-102L precision potentiometer equipped with a 3D printed pulley. For our development and test we used a pulley of 33 mm in diameter made of PLA plastic whose design files (.obj and .skp) are reported in the supplemental. This allow to have a

measurement range of 1023 mm and an associate accuracy of ±1mm (nominal accuracy of 0.1%). Such range can be variated using a pulley with different diameter. A change in range results in a change in accuracy and resolution. The pulley was printed using a 3D PRN LAB54 printer and a PLA filament of 1.75 mm in diameter (specific density 1488 $Kg/m^3$). The filament was extruded at 210°C with a velocity of 30 mm/s.

The system is powered using a 50 w solar panel and 12V, 12Ah battery. Since the Arduino UNO has a broad power input

range (recommended 7-12V) and in order to avoid overheating of the board connected to the use of a 12V power input voltage, a DC-DC converter is used to stabilize power voltage to 7.2V. The processing and storage module, the onboard operating temperature sensor and the linear position transducer are housed into a 15 x 10 x 7 cm waterproof box that, together with the power system, is housed into a second larger 40 x 30 x 15 cm waterproof box. Such modular structure is used to protect both electronic and mechanic components from environmental condition and allow to use very short cables

that, as well as the modular structure, prevent the system from thermal drift.

The code for the extensometer has been developed and compiled using the Arduino IDE environment and open source code-string available online. The logic of the code is reported in figure 2 and the code is reported in the supplemental as well as the sensors wiring schematics. The final cost of the extensometer was around 200€ including additional installation equipment (see Table 1; e.g. rebars, wire, screw etc…). Additionally, even if it has been developed specifically for earth

flows it can be used for all of the type of landslide that moves along well defined lateral slip surface and with specific improvements it can be installed also in different position like at the head of a landslide.

## 3 Installation and testing at the Pietrafitta earth flow

We test the extensometer performance at the Pietrafitta earth flow (Fig. 3a) in the Apennine mountains of southern Italy (Campania Region, Benevento Province). Since 2006, this earth flow has been periodically active exhibiting an alternation

of slow persistent movements and rapid movement especially localized at the toe of the flow. We chose this earth flow because it is actively moving, its movement occurs largely along lateral well-defined shear surface (e.g. Gomberg et al., 1995) and is monitored using both discrete GPS surveys and nearly continuous rTPS measurements. We installed our low-cost Arduino-based extensometer along the left flank of the earth flow Toe (Fig. 3b). The installation was completed using a 2.5 m long wire, supported by several rebars, that forms a rail parallel to the strike slip fault materializing the left flank of the

flow. In this way, the extensometer is dragged/moves along the flank registering the cumulative displacement (scheme is shown in figure 3c) every 30 seconds. We used available displacement data to make a comparative analysis of the monitored displacement. To compare displacement measured with different systems, we installed a GPS antenna screw mounting and a rTPS target on the wire extensometer (Fig. 3b). Raw data measured with our monitoring systems are shown in the graph of

figure 3d. Especially, the earth flow toe moved of approximately 1 m in 6 days and 6 hours. The average velocity calculated on the basis of these data was of ~6.6 mm/h. In this part of the flow, the movement was largely dominated by the horizontal component. This make possible the comparison between displacement measured by the extensometer and the horizontal component of the displacement vectors reconstructed with both the GPS surveys and the rTPS. The comparison of the results indicates that the total displacement measured by our extensometer was approximately the same measured combining GPS and rTPS surveys. Difference between total displacement measured by the combination of GPS surveys and rTPS and the extensometer was around 1.5 cm (1.5%). Additionally, the displacement time series reconstructed using rTPS data perfectly fits the extensometer time series for the first 4 days of rTPS monitoring, despite a slight thermal drift of the rTPS (see red curve of Fig. 3d). In the successive 2 days the degree of fit seems to decrease and affects the measured total displacement. Probably, this was also caused by the deformation of the rail induced by the tilting of the ground surface around the extensometer. Despite this drawback, the system exhibit a very high monitoring stability without visible thermal drift, also at operating temperature higher than 35°C.

## 4 Concluding remarks and possible future improvements

The Arduino based extensometer was developed to provide a low cost improvable platform for earth flow/landslide continuous monitoring. The prototype was developed on the basis of the Arduino Uno board and integrated a datalogging RTC shield and an operating temperature sensor. It was equipped with a high precision position transducer that in the test configuration offers a measuring range of 1023 mm and an associate accuracy of ±1mm. Field test indicates a high reliability of the measurement and the importance of the rail setup. Especially, for horizontal displacement monitoring it is important to consider the topography of the surface of the flow and possible surface deformation caused by movement. In this way, periodic inspection of the system needs to be planned. Major advantages of the system are: i) the very low cost, ii) the presence of integrated datalogger, iii) the potential to be integrated with additional sensors, iv) the possibility to be used for different type of landslides.

Even though our test indicates the ability of the system to work in real field conditions providing reliable data, we have to consider that our test was very short and in only 6 days our extensometer reaches the maximum measurable displacement (average velocity of 6.6 mm/h). Thus, for very low velocity and very long monitoring period it might be useful to use a 12-bit Arduino DUE board that permit to increase 4 times the resolution and/or the range. The system can be integrated with a data transmission shield that allow near real time data transmission and has the potential to be used in landslide emergency scenarios. To further reduce the cost of the device it would be possible to use the Arduino Pro Mini board that is cheaper than the Arduino UNO and has a lower power consumption that allow to choose also a cheaper power system and smaller housing boxes. Additionally, we have planned to replace the ABS plastic pulley with an aluminium pulley that should ensure a higher durability. This change might increase the cost of the system.

## Acknowledgement

We thank to Rolf Hut and an anonymous reviewer for their constructive review of the paper. We thank Remo Pedace of 3D MAKERS (Avellino, Italy) for his assistance in printing the pulley.

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

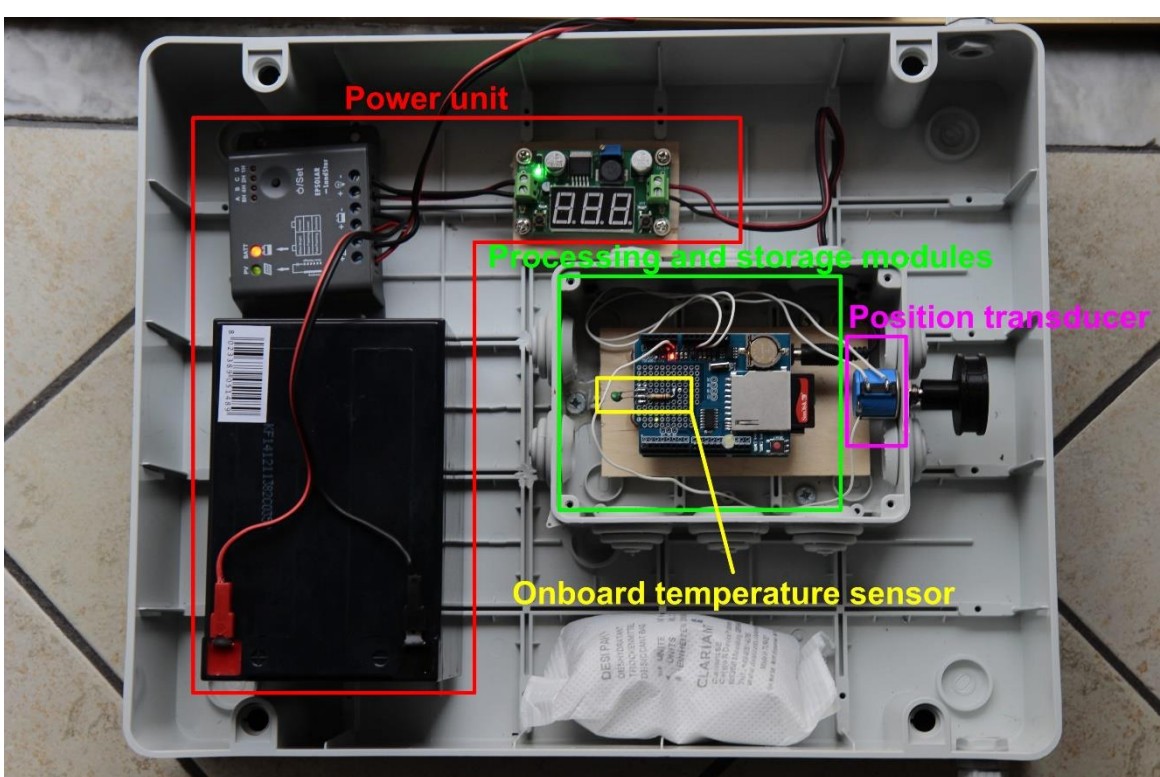

**Figure 1: The Arduino based extensometer after the assemblage. Major electronic components are labeled.**

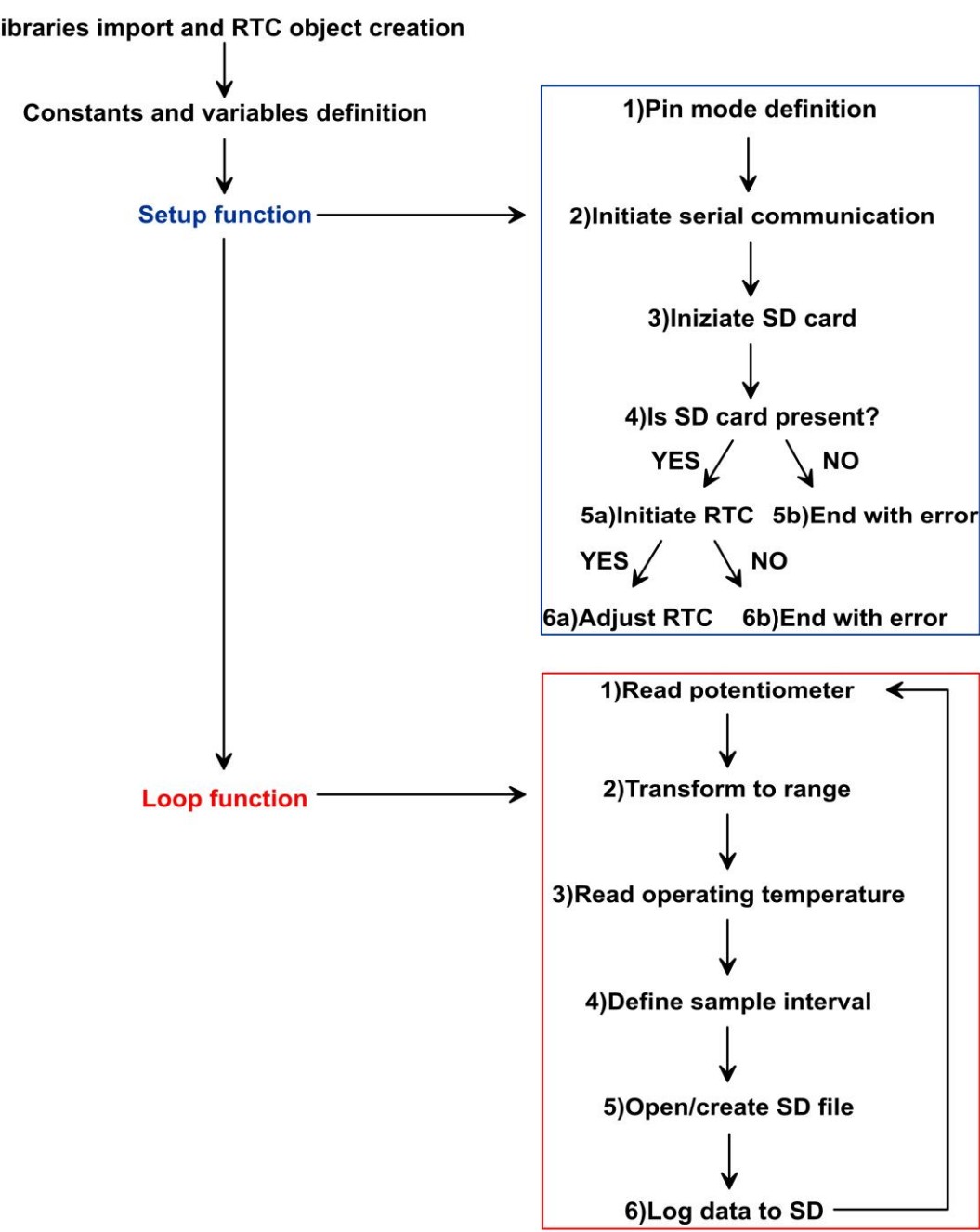

**Figure 2: Flow chart showing acquisition and storage logic.**

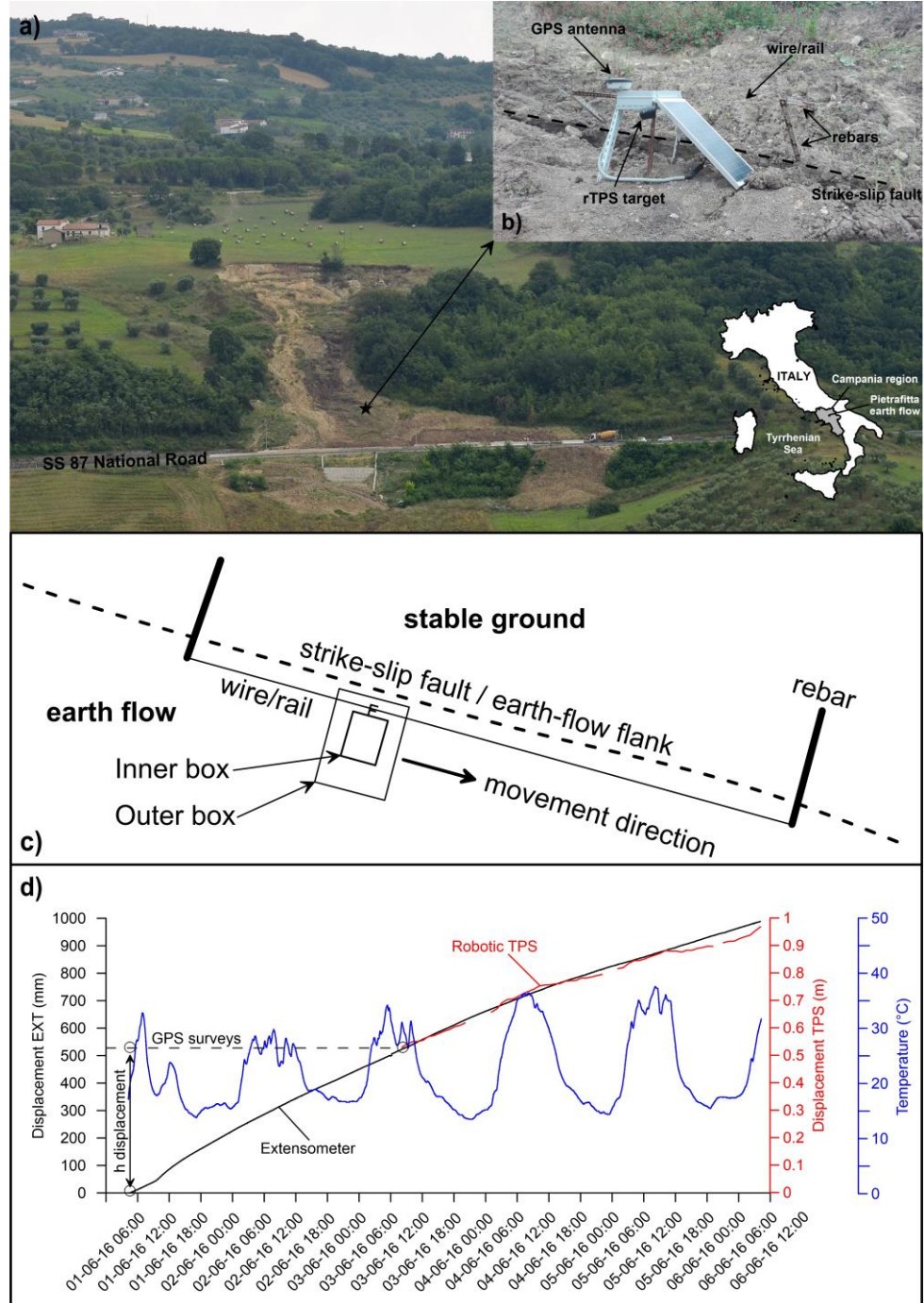

**Figure 3: a)** The Pietrafitta earth flow in the Appenine mountains of southern Italy, Campania region. The black star indicates position of the extensometer during field test. **b)** Installation configuration and monitoring equipment for comparative analysis. **c)** Installation scheme. **d)** Results of displacement monitoring with the extensometer and comparison with GPS derived and rTPS results. Black circles indicate error associated with GPS surveys. Operating temperature measured by the extensometer is also shown.

| Components* | Cost** (€) | Seller |
|---|---|---|
| Arduino UNO board | 22.0 | Online |
| Datalogging shield | 10.0 | Online |
| Precision potentiometer | 18.0 | Online |
| 3D pulley printing | 5.0 | Local |
| 10k thermistor + 10k resistor | 2.0 | Local |
| Led | 0.5 | Local |
| RTC coin cell battery | 2.5 | Local |
| DC-DC converter | 10.0 | Online |
| 50W Solar panel + 12Ah battery + 5A regulator | 60.0 | Online |
| Inner box | 8.0 | Local |
| Outer box + cable glands | 36.0 | Local |
| Cables + connector | 5.0 | Local |
| Wood panel (40x50 cm) | 7.0 | Local |
| Glue + rivets | 10.0 | Local |
| Rebars + screws | 25.0 | Local |
| Extensometer cable (steel fishing line) | 3.0 | Local |
| **Total** | **224.0** | |

*Minor components like board supports are not considered in the list because were already available.

**Online cost does not include shipping fees. Total shipping fees are around 35€.

**Table 1: Individual cost and seller type of each component of the monitoring system.**