# Peer review of "Brief Communication: low Arduino®-based wire A cost extensometer for earth flow monitoring"

_Natural Hazards and Earth System Sciences, 2017_

## Referee Comment (RC1) · R. Hut (Referee) · 22 Feb 2017

I have read the brief communication with great interest. I believe that a low cost device to measure landslide dynamics on relevant time and spatial scales is useful to the readership of NHESS. I will focus my review on the device itself and not on the landslide fieldwork since, although low cost devices used in geoscience are my expertise, landslides are not.

Interesting though the device presented is, the goal of brief communications about novel devices should, in my opinion, be to allow fellow geoscientists to successfully gauge whether the presented device is useful in their own research and if so, be able to successfully acquire or build and subsequently use it. In its current presentation the brief communication presented by the authors fails in this regard.

I would like to ask the authors to provide the following details, allowing the readership of NHESS to be able to reproduce the device:

1. Provide the code that runs on the Arduino. Figure 1 is a good start explaining the idea behind the code, but the authors should provide the actual code. Both for the readership as well as for the reviewers to be able to validate the work. In the open source spirit of Arduino, it seems that not providing the actual code to run the device was an oversight. The code can either be provided as supporting material to the paper, or it can be uploaded to Github and provided with a DOI through zenodo.org. This allows the authors to cite the code in this article.

2. Provide the design files needed to print the pulley that is attached to the potentiometer and provide details on both the specific 3D printer (including settings) and the material used in 3D printing.

3. Provide a Bill of Materials (BOM) for all the components that are used in building the device, including, if available, the (online) location where the authors purchased the component.

4. In Figure 1, I can spot a reference resistor being used with the thermistor. From the figure, I deduce that this is a 10kOhm resistor, but I'd like the authors to specify this. Also, please provide a schematic on how the thermistor and resistor are wired with respect to the ports (pins) of the Arduino.

Furthermore, to able to gauge whether the device is useful in fieldworks of the readership of NHESS, I would like the authors to clarify the following points:

1. The temperature is measured along with the displacement, however no details on the calibration used to infer temperature from recorded voltages are given: did the authors calibrate the thermistor themselves, or did they use factory provided

calibration constants. If so, how accurate are these? Can the temperature drift that is only gauged visibly be quantified?

2. The authors made a lot of choices in designing the device that are not made explicit in the paper. While most of these choices could be justified by arguments as simple as "familiarity with this type of component", it would be beneficial for the readership if those were made explicit in the paper. For example:

    (a) There are a lot of different SD loggers available for the Arduino eco-system. Why did the authors choose the one they did?

    (b) The Arduino Uno has a broad power input range (recommended 7-12V, limit 6-20V). There may be valid reasons to use a DC-DC converter between the 12V power source and the Arduino, but those are not mentioned. Please elaborate on this.

    (c) On the topic of power: for most fieldworks power consumption is an issue and the Arduino Uno uses considerably more power than some of it relatives, such as the Arduino Pro Mini or the third party Alog (http://northernwidget.com/alog/). The same holds true for the SD logger compared to some other SD loggers such as the Sparkfun OpenLog. Since the authors focus on the low cost of their device, and since I guess that the solar panel and battery used would contribute significantly to this cost, any power saving choices would be interesting and thus need to be elaborated on. Again: familiarity with the Uno could be all the reason the authors had, but I'd like the authors to specify this.

Finally, I have a rather unusual request: if possible, I'd like to ask the authors to provide the readership with a step-by-step guide on how to build the device. This could be supporting material to the article, or it could be a guide uploaded on instructables.com and cited in the article. This is an unusual request and may be outside the scope of

NHESS. I leave it to the editor to decide if he thinks this last request would further enhance the paper.

Hoping to see an updated version of the article,

Dr. ir. Rolf Hut

---

## Author Comment (AC1) · 7 Mar 2017

March 07, 2017

Dear Reviewer,

Thank you for proving comments to our manuscript. Below, we report the response to individual comments. The line numbers refer to the submitted manuscript.

Best regards,

Luigi Guerriero email: luigi.guerriero@unisannio.it

R. Hut (Referee) r.w.hut@tudelft.nl I have read the brief communication with great interest. I believe that a low cost device to measure landslide dynamics on relevant time and spatial scales is useful

to the readership of NHESS. I will focus my review on the device itself and not on the landslide fieldwork since, although low cost devices used in geoscience are my expertise, landslides are not. Interesting though the device presented is, the goal of brief communications about novel devices should, in my opinion, be to allow fellow geoscientists to successfully gauge whether the presented device is useful in their own research and if so, be able to successfully acquire or build and subsequently use it. In its current presentation the brief communication presented by the authors fails in this regard. I would like to ask the authors to provide the following details, allowing the readership of NHESS to be able to reproduce the device: 1. Provide the code that runs on the Arduino. Figure 1 is a good start explaining the idea behind the code, but the authors should provide the actual code. Both for the readership as well as for the reviewers to be able to validate the work. In the open source spirit of Arduino, it seems that not providing the actual code to run the device was an oversight. The code can either be provided as supporting material to the paper, or it can be uploaded to Github and provided with a DOI through zenodo.org. This allows the authors to cite the code in this article. 1 Response. We agree with the reviewer. Thus, we will provide the code as supporting material to the paper. 2. Provide the design files needed to print the pulley that is attached to the potentiometer and provide details on both the specific 3D printer (including settings) and the material used in 3D printing. 2 Response. We agree with the reviewer. Thus, we will provide the file needed to print the pulley as sketchup project and/or .obj files (as supplement) and we will add specific details about the material used in 3D printing and the 3D printer model and settings. Especially, we would modify the sentence of page 2, lines 28 and 29, in "For our development and test we used a pulley of 33 mm in diameter made of PLA plastic" and, in the same paragraph we will add the sentences "The pulley was printed using a 3D PRN LAB54 printer and a PLA filament of 1.75 mm in diameter (specific density 1488 Kg/m3). The filament was extruded at 210°C with a velocity of 30 mm/s". Among the future improvement, we plan to use an aluminum pulley. We will add two specific sentences in the Concluding remarks and possible future

improvements: "Additionally, we have planned to replace the ABS plastic pulley with an aluminum pulley that should ensure a higher durability. This change might increase the cost of the system". 3. Provide a Bill of Materials (BOM) for all the components that are used in building the device, including, if available, the (online) location where the authors purchased the component. 3 Response. In this case we disagree with the reviewer. Especially, since it is very easy to find online the components used for our extensometer and in order to avoid to publish documents that might be considered as advertisements, we prefer to do not show bills of material and /or indicate online shops. Additionally, several components were purchased to local specialized seller that has not an online shop. Minor components were already available in our laboratory (e.g. board supports). Having say that, in order to provide a general indication about component cost and seller type (local or online) we will add a specific table in the text. 4. In Figure 1, I can spot a reference resistor being used with the thermistor. From the figure, I deduce that this is a 10kOhm resistor, but I'd like the authors to specify this. Also, please provide a schematic on how the thermistor and resistor are wired with respect to the ports (pins) of the Arduino. 4 Response. We agree with the reviewer. Thus, we will add these details in the text. We would modify the sentence of page 2, lines 26 and 27, in "It is installed with a reference resistor of 10K in the prototyping area of the logging shield (see wiring schematic in the code attached as supplemental)" and would add wiring schematics in the code. Furthermore, to able to gauge whether the device is useful in fieldworks of the readership of NHESS, I would like the authors to clarify the following points: 1. The temperature is measured along with the displacement, however no details on the calibration used to infer temperature from recorded voltages are given: did the authors calibrate the thermistor themselves, or did they use factory provided calibration constants. If so, how accurate are these? Can the temperature drift that is only gauged visibly be quantified? 1 Response. We agree with the reviewer. Thus, we will add briefly information about thermistor calibration and accuracy estimation in the text. We would add to page 2 several new sentences: "For our test, the thermistor was calibrated between -10

and 40°C and obtained Steinhart-Hart coefficients were used for temperature esti-
mation. Steinhart-Hart coefficients were calculated using SRS Thermistor Calculator
(http://www.thinksrs.com/downloads/programs/Therm%20Calc/NTCCalibrator/NTCcalculator.htm).
We estimated temperature error comparing thermistor measurement with a precision
thermometer (accuracy ±0.05°C) in laboratory controlled conditions. The RMSE
calculated on the basis of 40 observations (between -5 and 35°C) was of ∼1°C. As
reported in the proposed new text, we do not estimate calibration coefficient accuracy
but made a further comparison with a precision thermometer and calculated the RMSE
of our measuring system. About temperature drift, we do not observe any drift during
our field test. 2. The authors made a lot of choices in designing the device that are not
made explicit in the paper. While most of these choices could be justified by arguments
as simple as "familiarity with this type of component", it would be beneficial for the
readership if those were made explicit in the paper. For example: (a) There are a lot of
different SD loggers available for the Arduino eco-system. Why did the authors choose
the one they did? 2(a) Response. Simply we choose the cheapest equipped with RTC.
We would add these new sentences in the text: "To choose this shield, we considered
the presence of the RTC module and its cost. We choose the cheapest". (b) The
Arduino Uno has a broad power input range (recommended 7-12V, limit 6-20V). There
may be valid reasons to use a DC-DC converter between the 12V power source and
the Arduino, but those are not mentioned. Please elaborate on this. 2(b) Response.
We agree with the reviewer. Thus, we will add some details in the text. Especially, we
would add a new sentence to page 2: "Since the Arduino UNO has a broad power input
range (recommended 7-12V) and in order to avoid overheating of the board connected
to the use of a 12V power input voltage, a DC-DC converter is used to stabilize power
voltage to 7.2V". (c) On the topic of power: for most fieldworks power consumption is
an issue and the Arduino Uno uses considerably more power than some of it relatives,
such as the Arduino Pro Mini or the third party Alog (http://northernwidget.com/alog/).
The same holds true for the SD logger compared to some other SD loggers such as
the Sparkfun OpenLog. Since the authors focus on the low cost of their device, and

since I guess that the solar panel and battery used would contribute significantly to this cost, any power saving choices would be interesting and thus need to be elaborated on. Again: familiarity with the Uno could be all the reason the authors had, but I'd like the authors to specify this. 2(c) Response. We agree with the reviewer. We will add a specific sentence in the Concluding remarks and possible future improvements section: "To further reduce the cost of the device it would be possible to use the Arduino Pro Mini board that is cheaper than the Arduino UNO and has a lower power consumption that allow to choose also a cheaper power system and smaller housing boxes". 3. Finally, I have a rather unusual request: if possible, I'd like to ask the authors to provide the readership with a step-by-step guide on how to build the device. This could be supporting material to the article, or it could be a guide uploaded on instructables.com and cited in the article. This is an unusual request and may be outside the scope of NHESS. I leave it to the editor to decide if he thinks this last request would further enhance the paper. 3 Response. Since our device has a simple structure and is formed by a pretty low number of components we would not provide a step by step guide. However, if the editor thinks that this might further enhance the paper, we will provide a simplified photographic guide to build it.

---

## Referee Comment (RC2) · Anonymous Referee #2 · 16 Mar 2017

+ One limitation of this work is that it cannot monitor the displacement in the real time manner. The real time measurement will be useful for reducing the risk. In fact, integrating the RF module for Arduino board is not a difficult task. + The author should show what the value of sampling time. + What is the power scheme which is designed for this device in order to save the power. + The device shown in Fig. 1 is not suitable to use in a long time with the different weather conditions. + Author should discuss about the associate accuracy of $\pm 1$ mm, is this from the datasheet or the calibration process.

---

## Short Comment (SC1) · 20 Mar 2017

This brief communication regards a low cost Arduino-based wire-rail extensometer integrating a datalogger, a power system and multiple digital and analog inputs. This system is usable for very long monitoring period. The manuscript is well written, has important technical message, and should be of great interest to the readers.

---

## Author Comment (AC2) · 28 Mar 2017

March 28, 2017

Dear Reviewer,

Thank you for proving comments to our manuscript. Below, we report the response to individual comments. The line numbers refer to the submitted manuscript.

Best regards,

Luigi Guerriero

email: luigi.guerriero@unisannio.it

Anonymous Referee #2

+ One limitation of this work is that it cannot monitor the displacement in the real time manner. The real time measurement will be useful for reducing the risk. In fact, integrating the RF module for Arduino board is not a difficult task. + Response. We agree with the reviewer that the integration of a RF module would improve our extensometer extending its applicability. However, this task is planned for the near future (see page 4, lines 14-16) and does not affect the performance of the instrument.

+ The author should show what the value of sampling time. +Response. The sampling time is already indicated, see page 3, lines 18-19.

+ What is the power scheme which is designed for this device in order to save the power. +Response. In designing the power system, we considered the factory declared power consumption of each component used in our monitoring system. We will add this detail in the text.

+ The device shown in Fig. 1 is not suitable to use in a long time with the different weather conditions. +Response. We disagree with this comment. Figure 1 shows major electronic components of the device. In field conditions, the power system is protected by a waterproof box and the Arduino board, the datalogging shield and the sensors are protected by an additional inner waterproof box. These make them protected by severe weather conditions and suitable for long term monitoring.

+ Author should discuss about the associate accuracy of $\pm 1$ mm, is this from the datasheet or the calibration process. +Response. We derived this accuracy from sensor datasheet (accuracy of 0.1% of the range). We will add this detail in the text.

---

## Author Response (AR1)

May 08, 2017

Dear Editor,

Below, we report the response to individual comments. The line numbers refer to the submitted manuscript. The new text in the revised manuscript is red and the deleted text is strikethrough. Please consider our revised manuscript for publication in Natural Hazards and Earth System Sciences.

Best regards,

Luigi Guerriero

email: luigi.guerriero@unisannio.it

Response to R. Hut (Referee) comments:

I have read the brief communication with great interest. I believe that a low cost device to measure landslide dynamics on relevant time and spatial scales is useful to the readership of NHESS. I will focus my review on the device itself and not on the landslide fieldwork since, although low cost devices used in geoscience are my expertise, landslides are not.

Interesting though the device presented is, the goal of brief communications about novel devices should, in my opinion, be to allow fellow geoscientists to successfully gauge whether the presented device is useful in their own research and if so, be able to successfully acquire or build and subsequently use it. In its current presentation the brief communication presented by the authors fails in this regard. I would like to ask the authors to provide the following details, allowing the readership of NHESS to be able to reproduce the device:

1. Provide the code that runs on the Arduino. Figure 1 is a good start explaining the idea behind the code, but the authors should provide the actual code. Both for the readership as well as for the reviewers to be able to validate the work. In the open source spirit of Arduino, it seems that not providing the actual code to run the device was an oversight. The code can either be provided as supporting material to the paper, or it can be uploaded to Github and provided with a DOI through zenodo.org. This allows the authors to cite the code in this article.

*1 Response. Change made. We provided the code as supporting material to the paper.*

2. Provide the design files needed to print the pulley that is attached to the potentiometer and provide details on both the specific 3D printer (including settings) and the material used in 3D printing.

*2 Response. Change made. We provided the files needed to print the pulley as sketchup project and .obj files as supporting material. We added specific details about the material used in 3D printing and the 3D printer model and settings. Especially, we modified the sentence of page 2, lines 28 and 29, in "For our development and test we used a pulley of 33 mm in diameter made of PLA plastic" and, in the same paragraph we added the sentences "The pulley was printed using a 3D PRN LAB54 printer and a PLA filament of 1.75 mm in diameter (specific density 1488 Kg/m³). The filament was extruded at 210°C with a velocity of 30 mm/s". Additionally, since among the future improvement, we plan to use an aluminum pulley, we added two specific sentences at the end of the Concluding remarks and possible future improvements section: "Additionally, we have planned to replace the ABS plastic pulley with an aluminum pulley that should ensure a higher durability. This change might increase the cost of the system".*

3. Provide a Bill of Materials (BOM) for all the components that are used in building the device, including, if available, the (online) location where the authors purchased the component.

*3 Response. Change made. In order to provide a general indication about component cost and seller type (local or online) we added a specific table (Table 1) in the text.*

4. In Figure 1, I can spot a reference resistor being used with the thermistor. From the figure, I deduce that this is a 10kOhm resistor, but I'd like the authors to specify this. Also, please provide a schematic on how the thermistor and resistor are wired with respect to the ports (pins) of the Arduino.

*4 Response. Change made. We added these details in the text. We modified the sentence of page 2, lines 26 and 27, in "It is installed with a reference resistor of 10K in the prototyping area of the logging shield (see wiring schematic in the code attached as supplemental)" and added the wiring schematics in the code.*

Furthermore, to able to gauge whether the device is useful in fieldworks of the readership of NHESS, I would like the authors to clarify the following points:

5. The temperature is measured along with the displacement, however no details on the calibration used to infer temperature from recorded voltages are given: did the authors calibrate the thermistor themselves, or did they use factory provided calibration constants. If so, how accurate are these? Can the temperature drift that is only gauged visibly be quantified?

*5 Response. Change made. We added information about thermistor calibration and accuracy estimation in the text. Especially, we added several new sentences to page 2: "For our test, the thermistor was calibrated between -10 and 40°C and obtained Steinhart-Hart coefficients were used for temperature estimation. Steinhart-Hart coefficients were calculated using SRS Thermistor Calculator (http://www.thinksrs.com/downloads/programs/Therm%20Calc/NTCCalibrator/NTCcalculator.htm). We estimated temperature error comparing thermistor measurement with a precision thermometer (accuracy ±0.05°C) in laboratory controlled conditions. The RMSE calculated on the basis of 40 observations (between -5 and 35°C) was of ~1°C".*

*As reported in the new text, we did not estimate calibration coefficient accuracy but made a further comparison with a precision thermometer and calculated the RMSE of our measuring system. About temperature drift, we did not observe any drift during our field test.*

6. The authors made a lot of choices in designing the device that are not made explicit in the paper. While most of these choices could be justified by arguments as simple as "familiarity with this type of component", it would be beneficial for the readership if those were made explicit in the paper. For example:

(a) There are a lot of different SD loggers available for the Arduino eco-system. Why did the authors choose the one they did?

*(a) Response. Simply we choose the cheapest equipped with RTC. We added two new sentences in the text: "To choose this shield, we considered the presence of the RTC module and its cost. We choose the cheapest".*

(b) The Arduino Uno has a broad power input range (recommended 7-12V, limit 6-20V). There may be valid reasons to use a DC-DC converter between the 12V power source and the Arduino, but those are not mentioned. Please elaborate on this.

*(b) Response. Change made. We added a new sentence to page 2: "Since the Arduino UNO has a broad power input range (recommended 7-12V) and in order to avoid overheating of the board connected to the use of a 12V power input voltage, a DC-DC converter is used to stabilize power voltage to 7.2V".*

(c) On the topic of power: for most fieldworks power consumption is an issue and the Arduino Uno uses considerably more power than some of it relatives, such as the Arduino Pro Mini or the third party Alog (http://northernwidget.com/alog/). The same holds true for the SD logger compared to some other SD loggers such as the Sparkfun OpenLog. Since the authors focus on the low cost of their device, and since I guess that the solar panel and battery used would contribute significantly to this cost, any power saving choices would be interesting and thus need to be elaborated on. Again: familiarity with the Uno could be all the reason the authors had, but I'd like the authors to specify this.

*(c) Response. Change made. We added a specific sentence in the Concluding remarks and possible future improvements section: "To further reduce the cost of the device it would be possible to use the Arduino Pro Mini board that is cheaper than the Arduino UNO and has a lower power consumption that allow to choose also a cheaper power system and smaller housing boxes".*

7. Finally, I have a rather unusual request: if possible, I'd like to ask the authors to provide the readership with a step-by-step guide on how to build the device. This could be supporting material to the article, or it could be a guide uploaded on instructables.com and cited in the article. This is an unusual request and may be outside the scope of NHESS. I leave it to the editor to decide if he thinks this last request would further enhance the paper.

*3 Response. Partial change made. We provided a simplified photographic guide to build it as supporting material.*

Response to Anonymous Referee #2

1 One limitation of this work is that it cannot monitor the displacement in the real time manner. The real time measurement will be useful for reducing the risk. In fact, integrating the RF module for Arduino board is not a difficult task.

*1 Response. No change made. We agree with the reviewer that the integration of a RF module would improve our extensometer extending its applicability. However, this task is planned for the near future (see page 4, lines 14-16) and does not affect the performance of the instrument.*

2 The author should show what the value of sampling time.

*2 Response. The sampling time is already indicated, see page 3, lines 18-19.*

3 What is the power scheme which is designed for this device in order to save the power.

*3 Response. In designing the power system, we considered the factory declared power consumption of each component used in our monitoring system. We added this detail in the text.*

4 The device shown in Fig. 1 is not suitable to use in a long time with the different weather conditions.

*4 Response. Figure 1 shows major electronic components of the device. In field conditions, the power system is protected by a waterproof box and the Arduino board, the datalogging shield and the sensors are protected by an additional inner waterproof box. These make them protected by severe weather conditions and suitable for long term monitoring.*

5 Author should discuss about the associate accuracy of ±1 mm, is this from the datasheet or the calibration process.

*5 Response. We derived this accuracy from sensor datasheet (accuracy of 0.1% of the range). We added this detail in the text.*

---

## Author Response (AR2)

May 17, 2017

Dear Editor,

Below, we report the response to individual comments. The line numbers refer to the revised manuscript. The new text in the revised manuscript is red and the deleted text is strikethrough. Please consider our revised manuscript for publication in Natural Hazards and Earth System Sciences.

Best regards,

Luigi Guerriero
email: luigi.guerriero@unisannio.it

Response to R. Hut (Referee) comments:
The authors have to my satisfaction responded to my concerns. Barring a (very) minor correction I believe this article is ready to be accepted. I have to make the same remark as when I took up the review: my expertise is in low cost sensing, not in landslides, so I can only speak for the low cost sensing part of the work presented.

1)The small minor revision I have is that I would like the authors to add sentences in the main manuscript that point to the supporting material where relevant: when the code or the 3D printer is mentioned.
2)Change made. We modified two sentences in the revised manuscript. See page 3 (4 in this version), lines 3-4 and page 3 (4 in this version), lines 17-18, respectively.

best regards,

Rolf Hut

[revised manuscript text omitted]